# Cell competition is driven by Xrp1-mediated phosphorylation of eukaryotic initiation factor 2α

**Naotaka Ochi**[ID], **Mai Nakamura**[ID], **Rina Nagata**[ID], **Naoki Wakasa**[ID], **Ryosuke Nakano,
Tatsushi Igaki**[ID]*

Laboratory of Genetics, Graduate School of Biostudies, Kyoto University, Yoshida-Konoecho, Sakyo-ku,
Kyoto, Japan

* igaki.tatsushi.4s@kyoto-u.ac.jp

pgen.1009958

Howard Hughes Medical Institute, UNITED STATES

**Data Availability Statement:** All relevant data are
within the manuscript and its Supporting
Information files.

## Abstract

Cell competition is a context-dependent cell elimination via cell-cell interaction whereby unfit
cells ('losers') are eliminated from the tissue when confronted with fitter cells ('winners').
Despite extensive studies, the mechanism that drives loser's death and its physiological trig-
gers remained elusive. Here, through a genetic screen in *Drosophila*, we find that endoplas-
mic reticulum (ER) stress causes cell competition. Mechanistically, ER stress upregulates
the bZIP transcription factor Xrp1, which promotes phosphorylation of the eukaryotic trans-
lation initiation factor eIF2α via the kinase PERK, leading to cell elimination. Surprisingly,
our genetic data show that different cell competition triggers such as ribosomal protein
mutations or RNA helicase *Hel25E* mutations converge on upregulation of Xrp1, which
leads to phosphorylation of eIF2α and thus causes reduction in global protein synthesis and
apoptosis when confronted with wild-type cells. These findings not only uncover a core path-
way of cell competition but also open the way to understanding the physiological triggers of
cell competition.

## Author summary

Cell competition is an evolutionarily conserved quality control process that selectively
eliminates viable unfit cells ('losers') when coexisting with fitter cells ('winners') within a
growing tissue. A common feature of loser mutants is a reduction in protein synthesis
rate. Recent studies have shown that the bZIP transcription factor Xrp1 causes reduction
in protein synthesis in ribosomal protein-mutant losers, while the mechanism by which
Xrp1 reduces protein synthesis remained unknown. Here, through a genetic screen in
*Drosophila*, we find that mutations causing ER stress make cells losers when surrounded
by wild-type cells. Mechanistically, cells undergoing ER stress upregulate Xrp1, which
promotes phosphorylation of the eukaryotic translation initiation factor eIF2α via the
kinase PERK, thereby reducing global protein synthesis and inducing cell death. Crucially,
this mechanism also drives cell competition triggered by ribosomal protein or *Hel25E*
mutations, both of which cause reduction in protein synthesis rate. Our findings show

**Funding:** This work was supported in part by grants from the MEXT/JSPS KAKENHI (Grant Number 20H05320, 21H05284, and 21H05039) to T.I, Japan Agency for Medical Research and Development (Project for Elucidating and Controlling Mechanisms of Aging and Longevity; Grant Number 20gm5010001) to T.I, the Takeda Science Foundation to T.I, and the Naito Foundation to T.I. The funders had no role in study design, data collection and analysis, decision to publish, or preparation of the manuscript.

**Competing interests:** The authors have declared that no competing interests exist.

that cell competition is commonly driven by Xrp1-mediated phosphorylation of eIF2α and that ER stress or other environmental stresses activating integrated stress response signaling, which converge on the phosphorylation of eIF2α, could be a physiological trigger of cell competition.

## Introduction

Cell competition is an evolutionarily conserved quality control process that selectively eliminates viable unfit cells ('losers') when coexisting with fitter cells ('winners') within a growing tissue [1–3]. For instance, cells with heterozygous mutations in the ribosomal protein genes, called *Minute/+* (*M*/+) mutations, are viable on their own but are eliminated from *Drosophila* imaginal epithelium when surrounded by wild-type cells [4]. Similarly, *Drosophila* cells homozygously mutant for *Mahjong/VprBP* (*Mahj*) [5] or the RNA helicase *Helicase25E* (*Hel25E*) [6] are viable on their own but are eliminated by apoptosis when confronted with wild-type cells. Several other factors also cause cell competition in *Drosophila*, which include high-level expression of the oncogene Myc [7,8], elevated activity of JAK-STAT or Wnt/Wg signaling [9,10], inactivation of the Hippo pathway [11], and loss of apico-basal cell polarity [12,13]. However, the physiological triggers of cell competition have still remained unclear.

A genetic study in *Drosophila* has identified a basic leucine zipper domain (bZIP) transcription factor Xrp1 as essential for driving *M*/+ cell competition[14]. Xrp1 is upregulated in *M*/+ cell clones and contributes to their cell death [15] [16]. Intriguingly, Xrp1 upregulation is also required for *M*/+ cells to reduce protein synthesis levels [15]. However, the mechanisms of how Xrp1 reduces protein synthesis and how it contributes to loser's death remained unknown. We have recently found that, similarly to *M*/+ clones, loser clones such as *Hel25E* or *Mahj* mutant clones reduce protein synthesis levels compared to neighboring wild-type winners [6], which suggests a potential mechanistic link between the reduction of protein synthesis and induction of loser's death.

Under various stress conditions, cells adapt to the environment via activation of the integrated stress response (ISR) signaling, an evolutionarily conserved intracellular signaling network that restore cellular homeostasis [17,18]. These stresses include endoplasmic reticulum (ER) stress, nutrient deprivation, viral infection, and oxidative stress. The stresses are sensed by four specialized kinases (PERK, GCN2, PKR, and HRI) that converge on phosphorylation of the alpha subunit of the eukaryotic translation initiation factor 2 (eIF2α). For instance, upon accumulation of unfolded proteins in the ER, the ER-resident chaperone BiP/Hsc70-3 is released from PERK, leading to homodimerization and activation of the eIF2α kinase PERK. eIF2α phosphorylation results in a reduction in global protein synthesis, while allowing the translation of selected genes including activating transcription factor 4 (ATF4) [17,19]. Two of four eIF2α kinases, PERK and GCN2, are conserved in *Drosophila*, which are activated by ER stress and amino acid deprivation, respectively [20]. ER stress is induced by various intracellular factors that cause the accumulation of unfolded proteins in the ER, which leads to activation of the unfolded protein response (UPR) signaling to recover ER function. In the *Drosophila* UPR pathway, PERK phosphorylates eIF2α and thus inhibits global protein synthesis to decrease the burden of ER capacity, while the endoribonuclease inositol-requiring enzyme-1 (Ire1) activates the transcription factor Xbp1 via a specific mRNA splicing which leads to upregulation of various genes helping the recovery of ER function [19,21].

Here, through a genetic screen in *Drosophila*, we found that mutations that cause ER stress make cells to be losers of cell competition when surrounded by wild-type cells.

Mechanistically, ER stress, as well as other cell competition triggers such as *M/+* and *Hel25E* mutations, upregulate Xrp1, which promotes phosphorylation of eIF2α via PERK, thereby causing reduction in protein synthesis and induction of cell death. Our data suggest that ER stress or other environmental stresses activating ISR signaling, which converge on the phosphorylation of eIF2α, could be a physiological trigger of cell competition.

## Results

### ER stress causes cell competition

To investigate the mechanism and physiological triggers of cell competition, we conducted a large-scale ethyl methanesulfonate (EMS)-based genetic screen in *Drosophila* to isolate mutations that cause cell competition. Using the Flippase (FRT)/ Flp recognition target (FLP)-mediated genetic mosaic technique, we induced homozygous mutant clones in otherwise wild-type tissue in *Drosophila* eye-antennal discs and isolated 87 *cell competition–induction* (*ccp*) mutations, among ~12,500 mutant chromosomes, that are cell-viable but eliminated when confronted with wild-type cells [6] (to be published elsewhere). Interestingly, among these *ccp* mutants, our whole genome sequencing analysis identified several mutant lines that have mutations in the genes involved in ER stress. For instance, a complementation group *ccp-2*, *ccp18*, and *ccp-29* mutant lines carried nonsense or flame-shift mutations in *Elongator complex protein 3* (*Elp3*) (S1A Fig), a lysine lacetyltransferase whose loss-of-function was shown to activate UPR signaling [22]. *ccp-21* carried a nonsense mutation in *calreticulin* (S1A Fig), an ER chaperone whose mutation was shown to cause ER stress and UPR [23]. Furthermore, *ccp28* carried a nonsense mutation in the gene *wollknaeuel* (*wol*) (S1A Fig), a dolichyl-phosphate glucosyltransferase that is involved in N-linked protein glycosylation in the ER. Loss of *wol* causes disruption of glycosylation and thus prevents proper folding of proteins in the ER, leading to activation of UPR signaling [24]. Given the clear cell competition phenotype, we focused our subsequent analyses on *ccp28* mutant.

 *ccp28* mutant clones (white) generated in the eye discs were eliminated during development when surrounded by wild-type clones (red) (Fig 1A and 1B), while eyes entirely mutant for *ccp28* developed into almost normal size (Fig 1C), indicating a context-dependent elimination of *ccp28* cells when confronted with wild-type cells. The elimination of *ccp28* clones from the eye disc was suppressed by overexpression of the caspase inhibitor p35 (Fig 1D, 1E, and 1F, quantified in Fig 1H). In addition, *ccp28* mutant clones induced cell death specifically at the boundaries between mutant and wild-type clones both in the eye discs and wing discs (Figs 1I, 1J, 1J', and S1B, quantified in Fig 1K). A rescue experiment in which Wol was overexpressed in *ccp28* mutant clones strongly suppressed their elimination, while Wol overexpression alone did not affect tissue growth (Fig 1G, quantified in Fig 1H). These data indicate that clones of cells mutant for *wol* are eliminated as losers of cell competition when confronted with wild-type cells.

 Consistent with the previous report [24], *wol* mutant clones induced specific mRNA splicing of Xbp1, a marker for ER stress [25], as visualized by the *Xbp1-GFP* reporter (Fig 1L and 1M). In addition, clones of other loser mutants, *ccp-2* (*Elp3*) or *ccp-21* (*calreticulin*), also elevated the *Xbp1-GFP* signal (S1C and S1D Fig). The slightly smaller size of *wol*$^{-/-}$ eyes (Fig 1C) could be due to a broadly increased ER stress in the entire eye disc (S1E Fig). Intriguingly, cell death was not significantly increased in the entire *wol*$^{-/-}$ eye discs (S1F Fig), suggesting that the smaller eye is due to a growth defect caused by elevated ER stress. Together, these data suggest that clones of cells causing ER stress are eliminated by cell competition when surrounded by wild-type cells.

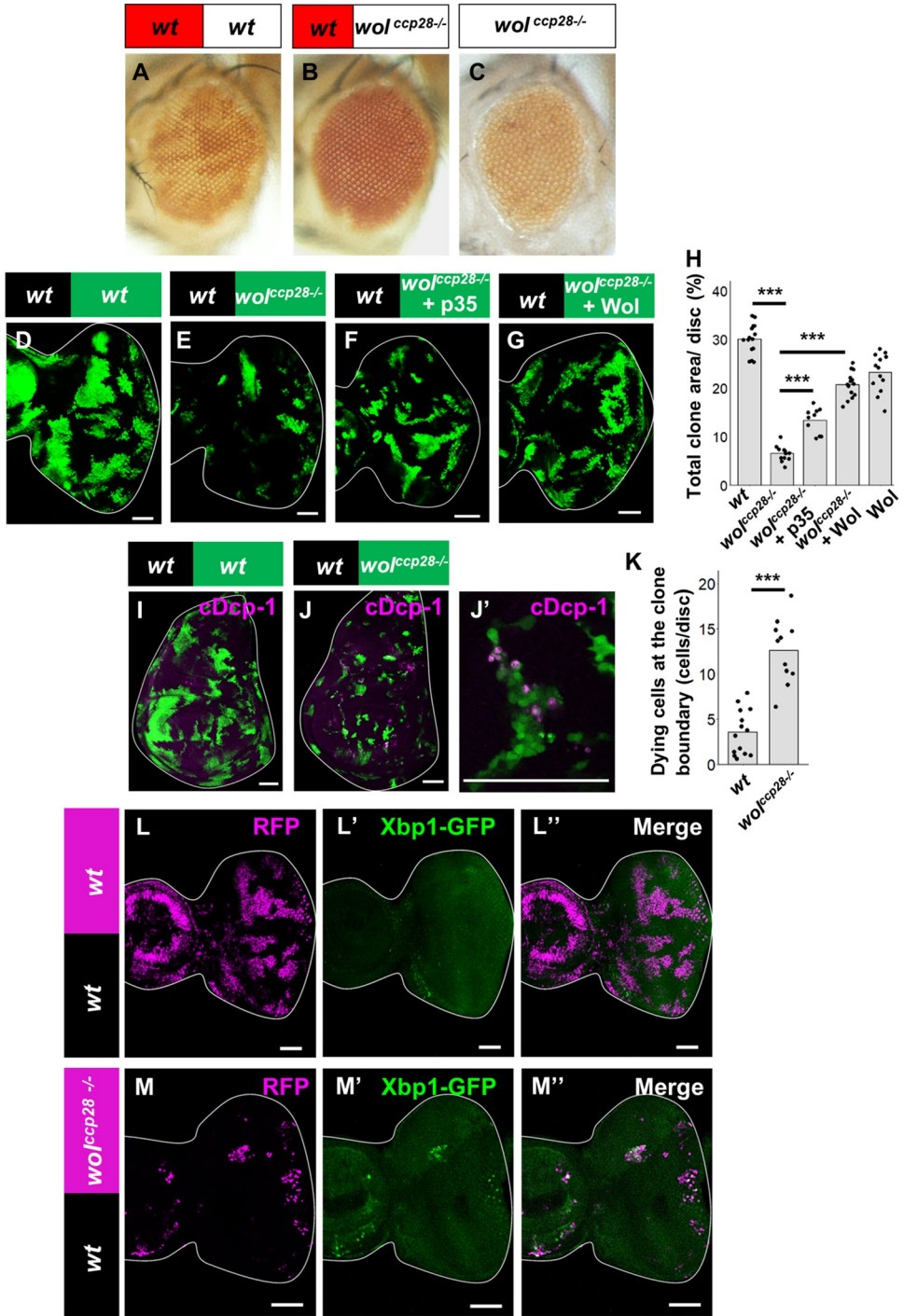

**Fig 1. ER stress causes cell competition.** (A and B) Adult eye bearing eyFLP-induced mosaics of wild-type (A) or *wol*<sup>ccp-28-/-</sup> (B) clones. (C) Adult eye bearing *wol*<sup>ccp-28-/-</sup> clones with surrounding wild-type tissue removed by GMR-hid and cell-lethal mutations. (D-G) Eye disc bearing eyFLP-induced MARCM clones of wild-type (D), *wol*<sup>ccp-28-/-</sup> (E), *wol*<sup>ccp-28-/-</sup> + UAS-p35 (F) cells or *wol*<sup>ccp-28-/-</sup> + UAS-wol (G) cells. (H) Quantification of the relative size of GFP-labeled clones shown in D-G. Error bars, SD; ***p<0.001 by Steel-Dwass test. (I-J') Wing disc bearing UbxFLP-induced MARCM clones of wild-type (I) or *wol*<sup>ccp-28-/-</sup> (J) stained with anti-cleaved Dcp-1. A magnified image of the inset in J is shown in J'. (K) Quantification of the number of dying cells at the boundary between wild-type and *wol*<sup>ccp-28-/-</sup> clones in the wing discs. Error bars, SD; ***p<0.001 by Welch's t-test. (L and M) Eye disc bearing eyFLP-induced MARCM clones of UAS-Xbp1-GFP (L) or *wol*<sup>ccp-28-/-</sup> + UAS-Xbp1-GFP (M) cells stained with anti-GFP. Scale bars, 50μm. See S1 Text for detailed genotypes.

## Cell competition is driven by phosphorylation of eIF2α

We next investigated the consequences of ER stress in loser cells. As expected, immunostaining analysis showed that most *wol* mutant clones elevate the phosphorylation of eIF2α (Fig 2A, quantified in Fig 2C), an indication of UPR activation [25]. The eIF2α phosphorylation in *wol* clones was canceled by knocking down the upstream kinase PERK (Fig 2B, quantified in Fig 2C). Strikingly, blocking PERK also significantly suppressed the elimination of *wol* clones (Fig 2B, compare to Fig 2A, quantified in Fig 2D), while PERK knockdown on its own did not affect tissue growth (Fig 2D). These effects were not observed by knockdown of another eIF2α kinase GCN2 [26] (S2A Fig), indicating that *wol* clones are eliminated via activation of the UPR pathway. Similar PERK-dependent elimination and elevated eIF2α phosphorylation were observed when homozygous clones of another *wol* mutant allele, *wol¹* [24], were analyzed. Given that eIF2α phosphorylation is known to reduce global protein synthesis, we analyzed protein synthesis rates using O-propargyl-puromycin (OPP) labeling assay [27]. Indeed, *wol* clones exhibited reduced protein synthesis compared to wild-type neighbors (Fig 2E), which was canceled by PERK knockdown (Fig 2F). These data indicate that PERK-mediated phosphorylation of eIF2α in *wol* clones causes both reduction in protein synthesis and induction of cell competition.

## ER stress upregulates Xrp1, which causes PERK-mediated eIF2α phosphorylation

Our data presented so far indicate an intriguing similarity between *M*/+ and ER stress-induced cell competition, as *M*/+ loser cells also exhibit reduced protein synthesis [15]. It has been shown that the upregulation of the transcription factor Xrp1 in *M*/+ clones is essential for the reduction in protein synthesis and their cell death [15]. We thus analyzed the role of Xrp1 in ER stress-induced cell competition and found interestingly that *Xrp1* expression was elevated in *wol* clones as visualized by the *Xrp1*-lacZ reporter [15](Fig 3A). In addition, knockdown of Xrp1 in *wol* clones significantly suppressed their elimination (Fig 3B, quantified in Fig 3C), while Xrp1 knockdown alone did not affect tissue growth (Fig 3C). Strikingly, Xrp1 knockdown also blocked the phosphorylation of eIF2α in *wol* clones (Fig 3B", compare to Fig 2A", quantified in Fig 3D). On the other hand, Xrp1 knockdown in *wol* clones did not suppress *Xbp1-GFP* signals (S3A Fig), indicating that ER stress still occurs in the absence of Xrp1. These data suggest that ER stress upregulates Xrp1, which causes phosphorylation of eIF2α and thus reduces protein synthesis. Indeed, Xrp1 knockdown in *wol* clones canceled the reduction in protein synthesis (Fig 3E). Moreover, overexpression of Xrp1 was sufficient to induce phosphorylation of eIF2α (Fig 3F, compare to S4A Fig, quantified in Fig 3H; S4C Fig). Furthermore, Xrp1-induced phosphorylation of eIF2α was canceled by PERK knockdown (Fig 3G, compare to S4B Fig, quantified in Fig 3H; S4D Fig). Together, these data indicate that ER stress upregulates Xrp1, which causes PERK-mediated phosphorylation of eIF2α, thereby causing reduction in protein synthesis and cell elimination.

## Xrp1-mediated phosphorylation of eIF2α commonly drives cell competition

Finally, we asked whether Xrp1-mediated activation of the UPR pathway also plays a critical role in cell competition triggered by other factors. To this end, we examined two different models of cell competition, the elimination of *RpL14*/+ mutant clones (*M*/+ cell competition) [28] and *Hel25E* mutant clones [6], both of which exhibit reduced protein synthesis compared to wild-type winners. Elimination of *RpL14*/+ clones from the wing disc was significantly

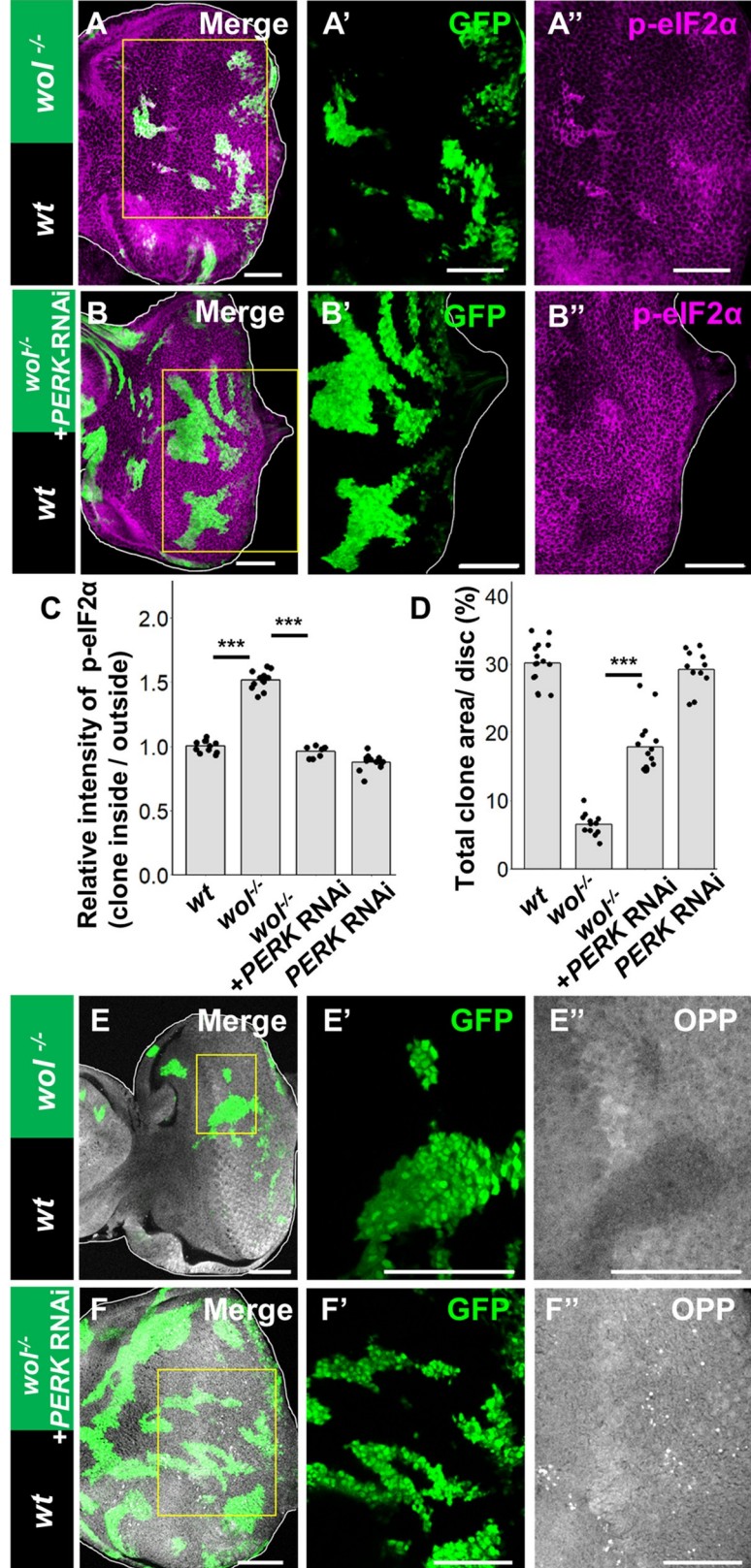

**Fig 2. Cell competition is driven by phosphorylation of eIF2α.** (A and B) Eye disc bearing eyFLP-induced MARCM clones of *wol*<sup>ccp-28-/-</sup> (A) or *wol*<sup>ccp-28-/-</sup> + PERK-RNAi (B) cells stained with anti-phosphorylated eIF2α. (C)

Quantification of the relative intensity of anti-phosphorylated eIF2α staining in clones shown in A and B. Error bars, SD; ***p<0.001 by Dunnett test. (D) Quantification of the relative size of GFP-labeled clones shown in A and B. Error bars, SD; ***p<0.001 by Welch's t-test. (E and F) Eye disc bearing eyFLP-induced MARCM clones of *wol*$^{ccp-28-/-}$ (E) or *wol*$^{ccp-28-/-}$+ PERK-RNAi (F) cells stained with OPP labeling. Scale bars, 50μm. See S1 Text for detailed genotypes.

suppressed by Xrp1 knockdown in *RpL14/+* clones as reported previously [15,16] (Fig 4A and 4B, quantified in Fig 4D) and notably, by PERK knockdown as well (Fig 4C, quantified in Fig 4D). Moreover, elimination of *Hel25E* clones from the eye disc was also strongly suppressed by knocking down Xrp1 or PERK in *Hel25E* clones (Fig 4E, 4F, and 4G, quantified in Fig 4H). These data indicate that the Xrp1-PERK pathway also plays a critical role in these cell competition models.

We then examined the signaling events occurring in these cell competition models. Consistent with the above data, *Xrp1* expression and eIF2α phosphorylation were both significantly upregulated in *RpL14/+* and *Hel25E* mutant clones (Fig 5A, 5B, 5C, and 5D, quantified in Fig 5E, 5F, 5G, and 5H). The induction of *Xrp1* expression in loser clones was not suppressed by PERK knockdown (S5A and S5B Fig, quantified in Fig 5E and 5G), indicating that Xrp1 acts upstream of PERK in these clones. On the other hand, the elevation of eIF2α phosphorylation was abolished when *PERK* or *Xrp1* was knocked down in these loser clones (Fig 5I, 5J, 5K, and 5L, quantified in Fig 5F and 5H). Together, these data indicate that different factors that induce cell competition commonly activate the Xrp1-PERK-eIF2α axis, which causes reduction in protein synthesis and induction of apoptosis in loser cells.

## Discussion

Our genetic data reveal that Xrp1-mediated phosphorylation of eIF2α plays a critical role in driving cell competition triggered by *M/+* mutation, *Hel25E* mutation, or ER stress. It has been shown that loser cells in *M/+* or *Hel25E*-induced cell competition commonly show lower protein synthesis levels compared to wild-type winners, but the mechanism by which they reduce protein synthesis remained unknown [15,29]. Our present data provide a mechanistic explanation that it is caused by global inhibition of translation by eIF2α phosphorylation.

Importantly, our data show that eIF2α phosphorylation is also required for the induction of loser's death. Similarly, recent studies have shown that *M/+* cells experience proteotoxic stress and thus induce phosphorylation of eIF2α, which acts as a driver of *M/+* cell competition [30–33]. Whether the global inhibition of protein synthesis or other downstream event(s) of eIF2α phosphorylation such as upregulation of UPR-activating transcription factor ATF4 is linked to their apoptosis is an outstanding important question. Notably, Xrp1 has been implicated to be a functional homolog of mammalian CHOP, a transcription factor that is induced by ATF4 [29]. Consistently, we found that overexpression of PERK leads to upregulation of Xrp1 expression (S6 Fig). In addition, recent studies have shown that overexpression of PERK or ATF4 upregulates Xrp1 [32,34]. These observations suggest that Xrp1 acts both upstream and downstream of the PERK-eIF2α axis in a positive feedback loop. The upstream Xrp1 may activate the PERK-eIF2α axis via upregulation of PERK expression [33]. Alternatively, Xrp1 upregulation may cause ER stress, which induces PERK activation. These are also important issues that should be addressed in the future studies.

It would also be important to clarify the mechanistic relationship between the Xrp1-PERK-eIF2α axis and other cell competition regulators so far reported, which include autophagy [6], Toll-related receptor signaling [28,35], Flower [36], and Azot [37]. In addition, it is crucial to understand in the future studies how Xrp1 is commonly upregulated by different cell competition triggers. Nonetheless, our current study identified a critical signaling axis that converge a

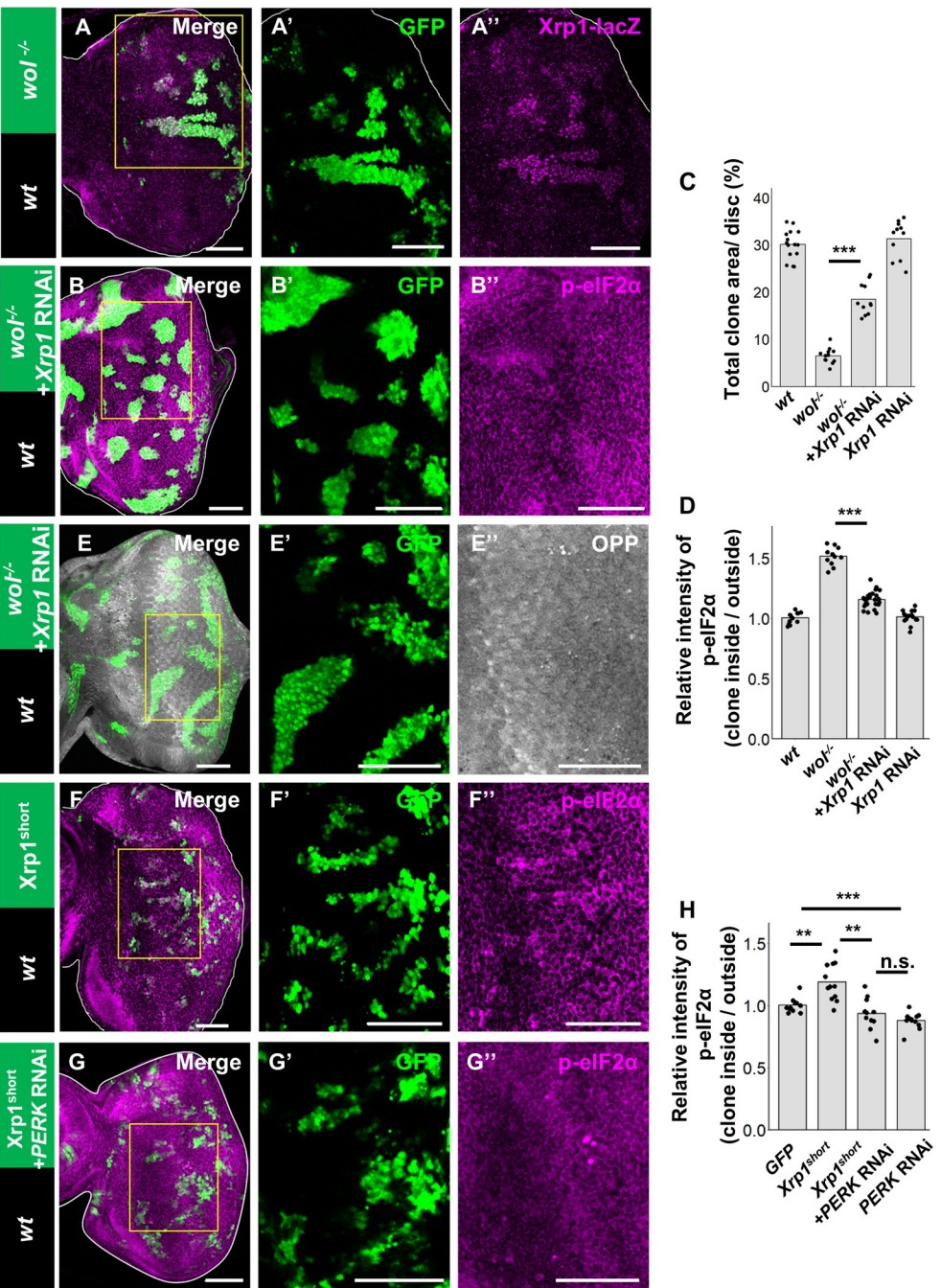

**Fig 3. ER stress upregulates Xrp1, which causes PERK-mediated eIF2α phosphorylation.** (A) *Xrp1-lacZ/+* eye disc bearing eyFLP-induced MARCM clones of *wol*<sup>ccp-28-/-</sup> stained with anti-β-gal. (B) Eye disc bearing eyFLP-induced MARCM clones of *wol*<sup>ccp-28-/-</sup> + Xrp1-RNAi cells stained with anti-phosphorylated eIF2α. (C) Quantification of the relative size of GFP-labeled clones shown in A and B. Error bars, SD; ***p<0.001 by Welch's t-test. (D) Quantification of the relative intensity of anti-phosphorylated eIF2α staining in clones shown in A and B. Error bars, SD; ***p<0.001 by Welch's t-test. (E) Eye disc bearing eyFLP-induced MARCM clones of *wol*<sup>ccp-28-/-</sup>+ Xrp1-RNAi cells stained with OPP labeling. (F and G) Eye disc bearing eyFLP-induced MARCM clones of UAS-Xrp1<sup>short</sup> (F) [46] or UAS-Xrp1<sup>short</sup> + PERK-RNAi (G) cells stained with anti-phosphorylated eIF2α. (H) Quantification of the intensity of anti-phosphorylated eIF2α staining in clones shown in F, G, and Fig 4B. Error bars, SD; ***p<0.001 by Steel-Dwass test. Scale bars, 50μm. See S1 Text for detailed genotypes.

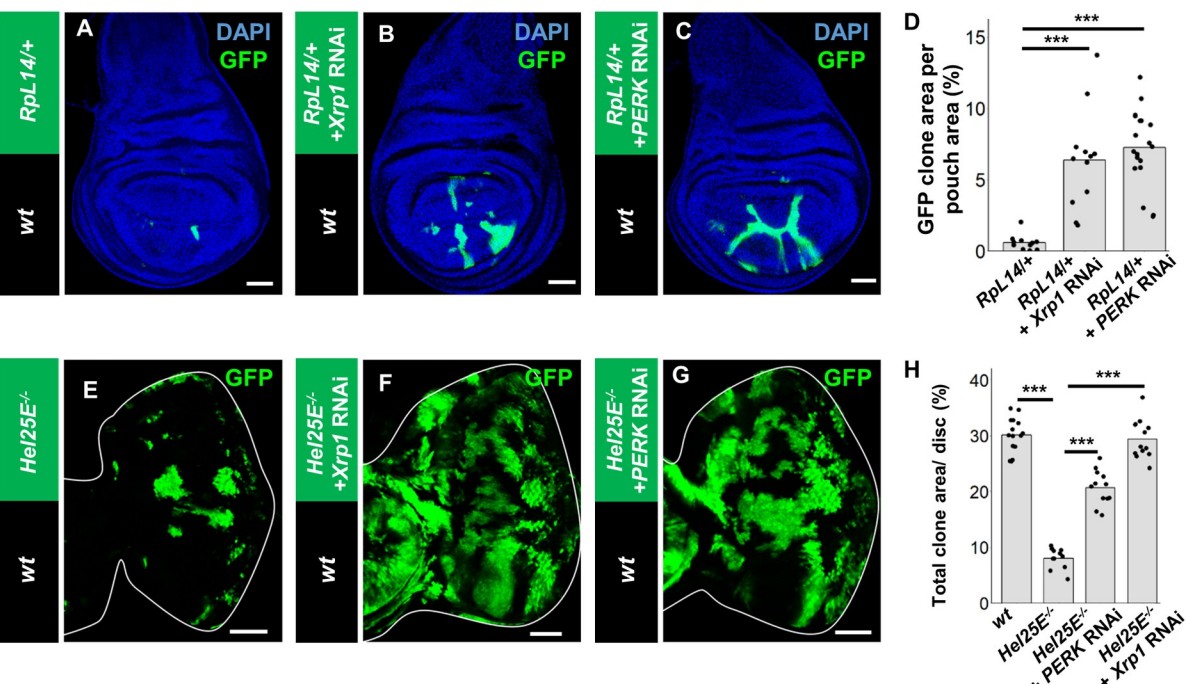

**Fig 4. The Xrp1-PERK pathway is commonly required for cell competition.** (A-C) Wing disc bearing hsFLP-induced GFP-labeled clones of *RpL14/+*, *salE*>GFP (A), *RpL14/+*, *salE*>GFP + *Xrp1-RNAi* (B), or *RpL14/+*, *salE*>GFP + *PERK-RNAi* (C) cells. (D) Quantification of the relative size of GFP-labeled clones shown in A-C. Error bars, SD; ***p<0.001 by Dunnett test. (E-G) Eye disc bearing eyFLP-induced MARCM clones of *Hel25E*$^{-/-}$ (E), *Hel25E*$^{-/-}$ + Xrp1-RNAi (F), or *Hel25E*$^{-/-}$ + PERK-RNAi (G) cells. (H) Quantification of the relative size of GFP-labeled clones shown in E-G. Error bars, SD; ***p<0.001 by Steel-Dwass test. Scale bars, 50μm. See S1 Text for detailed genotypes.

variety of cellular stress signaling to a common cell competition pathway via upregulation of Xrp1.

While studies in *Drosophila* have uncovered several triggers of cell competition and the downstream molecules essential for cell elimination [1], the physiological triggers of cell competition within animals have remained unknown. Our genetic screen identified a series of mutations that cause ER stress as triggers of cell competition. ER stress is induced by the accumulation of unfolded or misfolded proteins in the ER via a variety of intracellular factors under both physiological and pathological conditions [38–41], leading to activation of the evolutionarily conserved PERK-eIF2α pathway [19,42]. The PERK-eIF2α pathway is also activated by the conserved ISR signaling triggered by cell extrinsic factors such as amino acid deprivation, glucose deprivation, hypoxia, and viral infection. Moreover, Xrp1 expression is induced by genotoxic stresses such as irradiation [43]. Thus, our finding that the Xrp1-PERK-eIF2α axis commonly drives cell competition has opened the way to understanding the physiological and pathological role of cell competition. Intriguingly, mutations in the fused in sarcoma (FUS) gene, which are linked to amyotrophic lateral sclerosis (ALS), cause ER stress [40] and the *Drosophila* FUS orthologue cabeza genetically interacts with Xrp1[41]. In addition, it has been shown that cell competition plays a role in neurodegenerative diseases [44,45], which are thought to be driven by ER stress. Further studies on the physiological and pathological regulations of Xrp1-PERK-eIF2α signaling would unveil the *in vivo* role of cell competition.

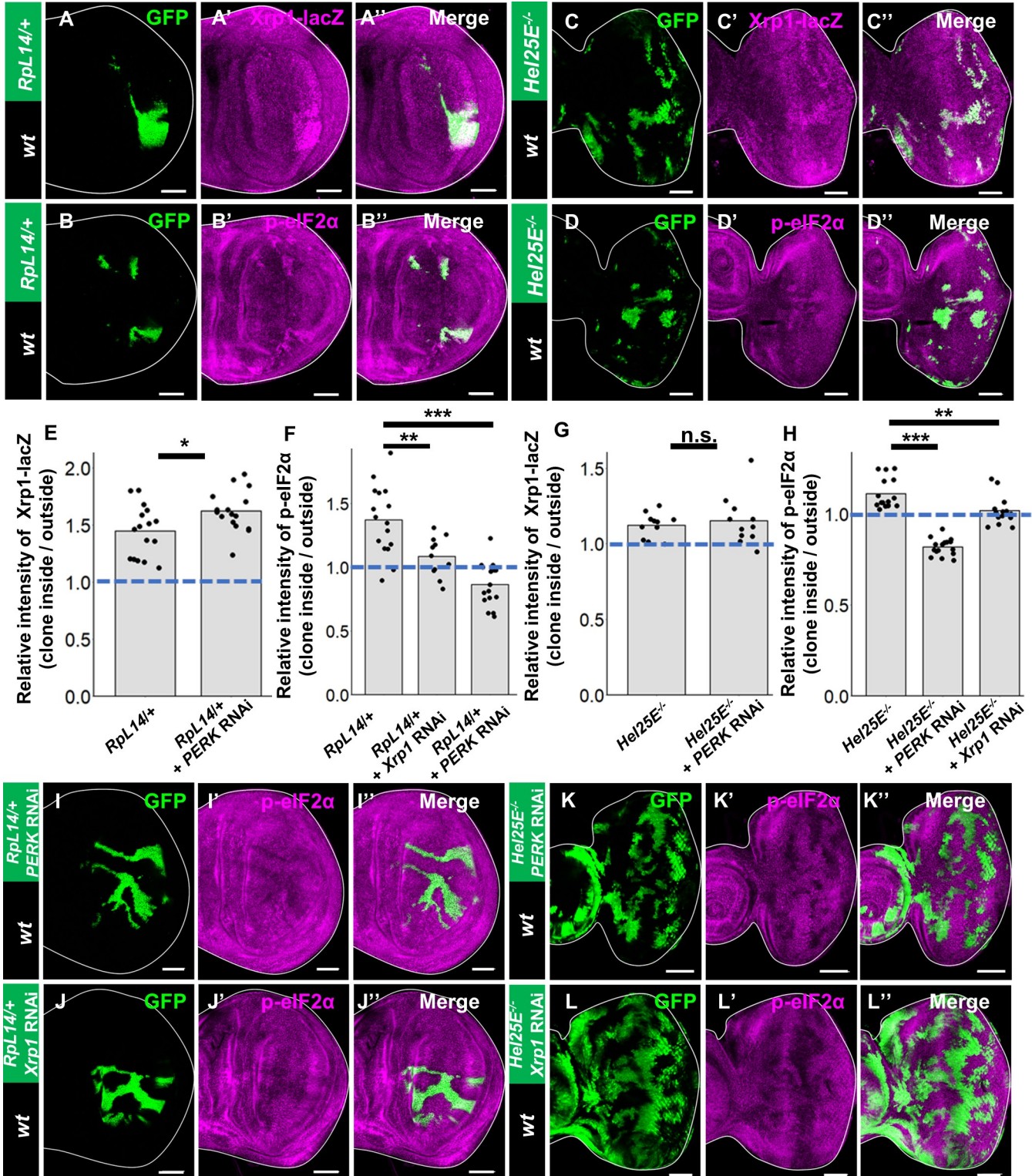

**Fig 5. Xrp1-mediated phosphorylation of eIF2α commonly drives cell competition.** (A-A") Xrp1-lacZ/+ background wing disc bearing hsFLP-induced GFP-labeled clones of RpL14/+, salE>GFP cells stained with anti-β-gal. (B-B") Wild-type background wing disc bearing hsFLP-induced GFP-labeled clones of RpL14/+, salE>GFP cells stained with anti-phosphorylated eIF2α. (C-C") Xrp1-lacZ/+ background eye disc bearing eyFLP-induced MARCM clones of Hel25E-/- cells stained with anti-β-gal. (D-D") Wild-type background eye disc bearing eyFLP-induced MARCM clones of Hel25E-/- cells stained with anti-phosphorylated eIF2α. (E) Quantification of the intensity of anti-β-gal staining in clones shown in A and S5.A. Error bars, SD; ***p<0.001 by Welch's t-test.

(F) Quantification of the intensity of anti-phosphorylated eIF2α staining in clones shown in B-D. Error bars, SD; ***p<0.001 by Dunnett test. (G) Quantification of the intensity of anti-β-gal staining in clones shown in E and S5.B. Error bars, SD; ***p<0.001 by Dunnett test. (H) Quantification of the intensity of anti-phosphorylated eIF2α staining in clones shown in F-H. Error bars, SD; ***p<0.001 by Dunnett test. Scale bars, 50μm. (I-I") Wild-type background wing disc bearing hsFLP-induced GFP-labeled clones of RpL14/+, salE>GFP + PERK RNAi cells stained with anti-phosphorylated eIF2α. (J-J") Wild-type background wing disc bearing hsFLP-induced GFP-labeled clones of RpL14/+, salE>GFP + Xrp1 RNAi cells stained with anti-phosphorylated eIF2α. (K-K") Wild-type background eye disc bearing eyFLP-induced MARCM clones of Hel25E-/- + PERK RNAi cells stained with anti-phosphorylated eIF2α. (L-L") Wild-type background eye disc bearing eyFLP-induced MARCM clones of Hel25E-/- + Xrp1 RNAi cells stained with anti-phosphorylated eIF2α. See S1 Text for detailed genotypes.

## Materials and methods

### Fly strain and generation of clones

*Drosophila melanogaster* strains were raised in vials containing a standard cornmeal-sucrose-yeast food, maintained at 25°C, unless otherwise stated. Sex of larvae dissected for most imaginal disc studies was not differentiated. Fluorescently-labeled mitotic clones (MARCM clones) were produced in larval imaginal discs using the following strains: tub-Gal80, FRT40A; eyFLP6, Act>y+>Gal4, UAS–GFP (40A tester), eyFLP1, UAS-Dicer2; Tub-Gal80, FRT40A; Act>y+>Gal4, UAS-GFP (40A Dicer2 tester), and Tub-Gal80, FRT40A; UAS-His2AmRFP, eyFLP6, Act>y+>Gal4 (40A RFP tester). Clones of *RpL14*/+ cells were generated using the following tester strains: hs-FLP, UAS-GFP::CD8; $M^{RpL14}$, salE>gRpL14>Gal4/TM6B (L. Johnston). Flies were allowed to lay eggs for 6 hours. Parental flies were removed, and larval progeny were heat-shocked 48 hours later at 37°C and analyzed using a fluorescence binocular or confocal microscope 72 hours later for clone measurements. Additional strains used are as follows: PERK-RNAi (NIG #GL00030), UAS-wol (FlyORF #F003019), UAS-dcr2 (Bloomington #24650), UAS-Xbp1-GFP (Bloomington #60731), Xrp1-RNAi (Bloomington #34521), Xrp1-lacZ (Bloomington #11569), UAS-Xrp1 (FlyORF #F000655), UAS-Xrp1$^{short}$ [46], Gcn2-RNAi (BDSC #35355), UAS-CD8-PARP-Vinus (Bloomington #65609), $wol^1$ [24] and UAS-GFP (BDSC).

### Genetic screen for *ccp* mutants

For a genetic screen for *ccp* mutants, male flies carrying an isogenized FRT40A chromosome (w/Y; FRT40A) were fed 25mM EMS and then mated to w; Kr/CyO females. Single F1 males of the genotype FRT40A∗/CyO were each crossed to 4–5 females of the genotype eyFLP1; Ubi-GFP, FRT40A/CyO (40A Ubi-GFP tester). Clone areas of white mutant cells compared to that of red-pigmented wild-type cells in the F2 eyes (non-CyO adult flies) were analyzed as the primary screen. In the secondary screen, wild-type cells surrounding mutant cells were removed using the GMR-hid, FRT40A, *l(2)Cl-L'*/CyO; ey-Gal4 UAS-FLP (40A CL tester) strain.

### Whole genome sequencing

Genomic DNA was extracted from 25–40 trans-heterozygous adult flies according to a standard phenol-chloroform method or NucleoSpin Tissue XS (MACHEREY-NAGEL). 150-bp paired-end sequencing at 30× coverage on Illumina Hi-Seq 4000 or Next Seq 500 was performed. The sequence reads were mapped to dm6 reference genome (UCSC version dm6) using BWA-MEM algorithm (Burrows-Wheeler Aligner) and calibrated based on Genome Analysis Tool-Kit (GATK) Best Practices (Broad Institute, http://www.broad.mit.edu). Genetic variants information was obtained by HaplotypeCaller in GATK. EMS-induced mutations were discriminated by the parental strain ('mutator') background and Drosophila Genetics Reference Panel (DGRP) database sequences. The mutations were then annotated and classified by SnpEff based on BDGP6.85 reference genome.

## Antibody staining

Wandering third instar larvae were dissected and fixed with 4% paraformaldehyde for 20 min at room temperature and blocked with 5% donkey serum and 0.1% Triton X-100 solution for 20 min. For immunostaining, samples were incubated at 4˚C overnight with primary antibodies, and then incubated with Alexa Fluor 405-, 488-, 546-, or 647-conjugated secondary antibodies (1:200, Thermo Fisher Scientific) for 2 hours at room temperature. The following primary antibodies were used: rabbit anti-cleaved *Drosophila* DCP1 (Cell Signaling Technology, 1:100), rabbit anti-phospho-eIF2α (Cell Signaling Technology, 1:100), rabbit anti-GFP (Nacalai Tesque, 1:250), chicken anti-β-gal (Abcam, 1:1000), anti-cleaved PARP antibody (Cell Signaling Technology, 1:200).

## Protein synthesis analysis

Nascent protein synthesis was analyzed using Click-iT Plus OPP Alexa Fluor 647 Protein Synthesis Assay Kit (Thermo Fisher Scientific). Wandering L3 larvae were dissected in Schneider's medium containing 5% FBS (Thermo Fisher Scientific), and incubated in 20μM OPP for 10min. After OPP incorporation, larvae were fixed in 4% paraformaldehyde for 20 min at room temperature and subsequently detected OPP by following manufacturer's manual.

## Image analysis

Confocal images were taken with SP8 Leica confocal microscope. Total clone area/disc are (%) was calculated per each discs using ImageJ software (NIH).

## Statistical analysis

R (ver. 4.1.1) was used for data plotting and statistical analyses. Raw data are shown as dot plot. The significance level was set to $p < 0.05$. Data were analyzed by Welch's t-test by single comparison. Data were analyzed by Steel-Dwass test or Dunnett test for multiple comparisons. Details of statistical evaluations and the numbers of samples were indicated in the figure legends. All data in bar graphs are expressed as mean ± SD. No statistical methods were used to predetermine sample size. All *n* numbers represent biological replicates. Each experiment was independently performed at least three times. All experiments were not randomized or blinded. In Fig 5E–5H, the relative intensity of Xrp1-lacZ or p-eIF2α signal in GFP-positive cells to GFP-negative cells was measured.

## Supporting information

**S1 Fig. ER stress underlies several different genetic contexts of cell competition.** (A) A list of isolated *ccp* mutants that cause ER stress.and the schematic representations of the general domain structures of Elp3, Calr, and Wol, with mutations detected by the whole genome sequencing. (B) Eye disc bearing eyFLP-induced MARCM clones of mCD8-PARP-Vinus-exprssing *wol*[ccp-28-/-] cells stained with anti-cleaved PARP (which detects caspase-activated dying cells). (C) Adult eye bearing eyFLP-induced mosaics of Elp3[ccp-2][-/-] clones (left panel) or Elp3[ccp-2][-/-] clones with surrounding wild-type tissue removed by GMR-hid and cell-lethal mutations (middle panel). Eye disc bearing eyFLP-induced MARCM clones (RFP) of Elp3[ccp-2][-/-] + UAS-Xbp1-GFP cells stained with anti-GFP (right panels). (D) Adult eye bearing eyFLP-induced mosaics of Calr [*ccp-21*][-/-] clones (left panel) or Calr [*ccp-21*][-/-] clones with surrounding wild-type tissue removed by GMR-hid and cell-lethal mutations (middle panel). Eye disc bearing eyFLP-induced MARCM clones (RFP) of Calr [*ccp-21*][-/-] + UAS-Xbp1-GFP cells stained with anti-GFP (right panels). Scale bars, 50μm. (E) Wild-type (left) or *wol*[ccp-28-/-]

(right) eye disc bearing UAS-Xbp1-GFP. In both tissues, wild-type or $wol^{ccp-28-/-}$ clones were induced in the eye disc and then surrounding wild-type tissue was removed by GMR-hid and cell-lethal mutations. Scale bars, 50μm. (F) Wild-type (left) or $wol^{ccp-28-/-}$ (middle) eye disc stained with anti-cleaved Dcp-1. In both tissues, wild-type or $wol^{ccp-28-/-}$ clones were induced in the eye disc and then surrounding wild-type tissue was removed by GMR-hid and cell-lethal mutations. Scale bars, 50μm. (right) Quantification of the number of dying cells in wild-type or $wol^{ccp-28-/-}$ eye discs. Error bars, SD; ***p<0.001 by Welch's t-test. See S1 Text for detailed genotypes.
(TIF)

**S2 Fig. GCN2 is required neither for eIF2α phosphorylation nor cell competition.** (A) Eye disc bearing eyFLP-induced MARCM clones of $wol^{ccp-28-/-}$ + Gcn2-RNAi cells stained with anti-phosphorylated eIF2α. Quantification of the relative size of GFP-labeled clones or relative intensity of anti-phosphorylated eIF2α staining shown in A. Error bars, SD; ***p<0.001 by Welch's t-test. Scale bars, 50μm. See S1 Text for detailed genotypes.
(TIF)

**S3 Fig. Xrp1 knockdown does not suppress ER stress caused by wol mutations.** (A) Eye disc bearing eyFLP-induced MARCM clones of $wol^{ccp-28-/-}$ + Xrp1-RNAi + UAS-Xbp1-GFP cells stained with anti-GFP. See S1 Text for detailed genotypes.
(TIF)

**S4 Fig. Xrp1 causes PERK-mediated phosphorylation of eIF2α.** (A) Eye disc bearing eyFLP-induced MARCM clones of UAS-GFP cells stained with anti-phosphorylated eIF2α. (B) Eye disc bearing eyFLP-induced MARCM clones of PERK RNAi cells stained with anti-phosphorylated eIF2α. (C) Wing disc overexpressing GFP, $Xrp1^{short}$ in the wing pouch by the *nub-Gal4* driver stained with anti-phosphorylated eIF2α. (D) Wing disc overexpressing GFP, $Xrp1^{FlyORF}$ (FlyORF: F000655) in the wing pouch by the *nub-Gal4* driver stained with anti-phosphorylated eIF2α. Scale bars, 50μm. See S1 Text for detailed genotypes.
(TIF)

**S5 Fig. Xrp1 expression is induced at the upstream of PERK.** (A) Xrp1-lacZ/+ background wing disc bearing hsFLP-induced GFP-labeled clones of RpL14/+, salE>GFP + PERK-RNAi cells stained with anti-β-gal. (B) Xrp1-lacZ/+ background eye disc bearing eyFLP-induced MARCM clones of $Hel25E^{-/-}$ + PERK RNAi cells stained with anti-β-gal. Scale bars, 50μm. See S1 Text for detailed genotypes.
(TIF)

**S6 Fig. PERK induces Xrp1 expression.** (A) Xrp1-lacZ/+ background wing disc overexpressing GFP in the wing pouch by the *nub-Gal4* driver stained with anti-β-gal. (B) Xrp1-lacZ/+ background wing disc overexpressing GFP, PERK in the wing pouch by the *nub-Gal4* driver stained with anti-β-gal. Scale bars, 50μm. See S1 Text for detailed genotypes.
(TIF)

**S1 Text. Detailed genotypes used in each figure.**
(DOCX)

## Acknowledgments

We thank Y. Noguchi, K. Baba, M. Tanaka, M. Koijima, M. Matsuoka, and K. Gomi for technical support, N. Baker, P. Leopold, M. Mannervik, HD. Ryoo, T. Uemura, the Bloomington *Drosophila* Stock Center, the National Institute of Genetics Stock Center (NIG-FLY), the

*Drosophila* Genomics and Genetic Resources (DGGR, Kyoto Stock Center), and the Vienna *Drosophila* Resource Center (VDRC) for fly stocks and reagents, and T. Kondo, Y Sando, and the NGS Core Facility of the Kyoto University Graduate Schools of Biostudies for the whole genome sequencing. We also thank Y. Sanaki, H. Kanda, and members of the Igaki laboratory for discussions.

## Author Contributions

**Conceptualization:** Naotaka Ochi, Tatsushi Igaki.

**Data curation:** Naotaka Ochi.

**Formal analysis:** Naotaka Ochi.

**Funding acquisition:** Tatsushi Igaki.

**Investigation:** Naotaka Ochi, Mai Nakamura, Rina Nagata, Naoki Wakasa, Ryosuke Nakano.

**Methodology:** Naotaka Ochi.

**Project administration:** Tatsushi Igaki.

**Resources:** Tatsushi Igaki.

**Supervision:** Tatsushi Igaki.

**Validation:** Naotaka Ochi.

**Visualization:** Tatsushi Igaki.

**Writing – original draft:** Naotaka Ochi, Tatsushi Igaki.

**Writing – review & editing:** Naotaka Ochi, Tatsushi Igaki.

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
