## [Decision Letter · Decision Letter 0]

20 May 2021

Dear Tatsushi,

Thank you very much for submitting your Research Article entitled 'Cell competition is driven by Xrp1-mediated phosphorylation of eukaryotic initiation factor 2α' to PLOS Genetics.

The manuscript was fully evaluated at the editorial level and by independent peer reviewers. The reviewers appreciated the attention to an important problem, but raised some  concerns about the current manuscript. Based on the reviews, we will not be able to accept this version of the manuscript, but we would be willing to review a revised version.

If you decide to revise the manuscript for further consideration at PLOS Genetics, please aim to resubmit within the next 60 days, unless it will take extra time to address the concerns of the reviewers, in which case we would appreciate an expected resubmission date by email to plosgenetics@plos.org.

[LINK]

We are sorry that we cannot be more positive about your manuscript at this stage. Please do not hesitate to contact us if you have any concerns or questions.

Yours sincerely,

Norbert Perrimon

Associate Editor

PLOS Genetics

Gregory P. Copenhaver

Editor-in-Chief

PLOS Genetics

Reviewer's Responses to Questions

**Comments to the Authors:**

Reviewer #1: Summary

In this manuscript, the authors show that mutations leading to ER stress are able to induce cell competition, which confirms previous studies that proteotoxic stresses is a driving force in Minute based cell competition (Baumgartner et al.,2021). In an EMS screen for cell competition factors they found genes involved in ER stress and the UPR. They then argue that these factors induce cell competition. Building on the connection they position Xrp1 upstream of the stress response. Increased expression of Xrp1 is suggested to increase of peIF2α levels, which in turn reduce protein translation.

The novelty of this paper derives from the description of the link between Xrp1 and PERK/ peIF2α. This connection will be of general interest. The report will certainly stimulate further work. Unfortunately, not all the conclusions are equally well substantiated. Additional experiments and/or extended discussion would be beneficial.

Major comments:

- The title of the paper is “Cell competition is driven by Xrp1-mediated phosphorylation of eukaryotic initiation factor 2α”. The implied “Xrp1-PERK-eIF2α axis” places Xrp1 upstream of PERK. While intriguing the provided evidence does not exclude that Xrp1 acts in a positive feedback loop. Based on the timing of the experiments can the authors exclude this alternative? How do the authors envisage that Xrp1 mediates the phosphorylation of eukaryotic initiation factor 2α? Xrp1 is a nuclear/nucleolar transcription factor PERK located on the ER membrane. Additional experiments and discussion should be provided or the title adjusted.

- The key experiments for their claim are presented in Figure 3. Xrp1 knock-down and overexpression are presented. In a wol-/- clone p-eIF2α levels are increased. This is reduced when Xrp1 is knocked down. Overexpression of Xrp1 increases p-eIF2α levels. The presented results are suggestive, but the quantification suggests a high degree of variability. An increased sample size would be helpful. Alternatively, it could help to express Xrp1 in, for example, the posterior compartment and compare the resulting p-eIF2α levels to those in the anterior compartment. There seems to be a significant variation in the background levels of p-eIF2α and this alternative approach may simplify the analysis and interpretation. Also, not all wol -/- clones seem to have elevated p-eIF2α (Figure 2a). This should be discussed.

- The data in figure 3F-H is not convincing, because only a weak upregulation of p-eIF2α can be seen upon Xrp1 overexpression. To exclude trivial explanations the authors should compare p-eIF2α expression in a control with a random UAS line (e.g. GFP or LacZ) and also the effect of PERKRNAi without Xrp1 overexpression. As Xrp1 overexpressing clones are very small, they could either induce bigger clones or show the effect of Xrp1 overexpression in compartments.

- They use two different UAS-lines of Xrp1 (Figure 3 and Figure S3), in which they name Xrp1 differently (Xrp1short and Xrp1). They should clarify the difference between the two lines and prove that both lines lead to the same phenotype.

- Data shown in figure 5 is not convincing. Their depicted Minute clone in 5a does not reflect the data of their graph in 5e. In Figure 5, they should show full discs containing several clones for all experiments. Showing additional close-ups of clones would also help. To corroborate the main message of the paper: the authors should quantify how the Xrp1 LacZ and peIF2α intensities are changing during cell competition when using Xrp1RNAi or PERKRNAi.

Minor comments:

- In places the text seems hastily assembled, which detracts unnecessarily from the otherwise thought-provoking data. The authors should carefully revise the text, as it is not updated on recent literature from their field and their citation style is not consistent. At the beginning of this year two studies have independently shown that proteotoxic stress is induced by a Minute situation and is essential for cell competition (Baumgartner et al., 2021, Recasens-Alvarez et al., 2021). Citing these is needed to properly place the claims in the context of the current literature.

- Authors should try to explain the link of Xrp1 and PERK in more detail. They should try to integrate what is known of endoplasmic reticulum unfolded protein responses and how this leads to an increased peIF2α. Which model would the authors propose how Xrp1 functions in their PERK-Xrp1- peIF2α model? What factors could be upstream of Xrp1? Blanco et al., proposed that the mammalian homolog of Xrp1 might be CHOP/ddit3. This would put Xrp1 below PERK. Discussing this would help place the claims in context.

- A further missing reference connecting ER stress and Xrp1 is Mallik et al.,2018. They reported a role for Xrp1 in the toxicity of the ALS-associated FUS orthologue caz mutant phenotype.

- Given the narrowness of the clones in Figure 1J/K, it is somewhat debatable to discuss apoptosis on the border vs middle of the clone. Hopefully alternative images could be provided.

Reviewer #2: Ochi, Igaki review:

In this very nice body of work, the authors carried out an EMS-based genetic screen using mitotic clones in the developing Drosophila eye to look for inducers of cell competition. They screened 12,500 mutant chromosomes and identified 87 mutations that caused cells to be competitively eliminated from mosaic eye (named ccp mutants). Genomic sequencing revealed mutations in three genes involved in ER stress: Elp3, calreticulin, and wollknaeuel (wol), involved in N-linked protein glycosylation in ER. Focusing on wol mutants, they found that mutants did not alter normal eye size (but see comments 1 and 2 below), but when with WT cells in mosaic eye discs wol mutant cells were eliminated via apoptosis (p35 prevented their elimination). Cell death was observed at clone boundaries, which is also seen in many contexts of cell competition.

The authors found that Xrp1 was upregulated in the wol clones, and p-eIF2a was increased, consistent with activation of the integrated stress response; accordingly, a reduction in OPP signal in the clones indicated that protein synthesis was downregulated. Clonal Xrp1 over-expression led to similar increases in p-eIF2a, which was suppressed by co-expression of PERK-RNAi. All of these results are consistent with activation of the ISR due to ER stress in the mutant cells. The authors then asked if ISR activation occurred in other competitive contexts by looking at mosaic discs containing clones of either Rpl14+/- cells, or Hel25E-/- cells. The mutant cells of both genotype are normally out competed by the WT cells, but when either Xrp1 or the ER stress kinase PERK was knocked down in the mutant clones, the cells were no longer eliminated. Altogether, their data suggest that the Xrp1-PERK-eIF2α axis is commonly activated in these different genetic contexts of cell competition, leading to a reduction in protein synthesis and induction of apoptosis in the loser cells. Intriguingly, GCN2, another ER stress sensing eIF2a kinase, was found not to be required, suggesting some interesting specificity.

The experiments in the paper are well done and nicely presented. The paper is also well written and for the most part the authors are appropriately circumspect in their conclusions (see comment #4 below). Work published very recently by others showed that Rp haploinsufficiency results in severe proteotoxic stress, activation of the ISR, and increased cell autonomous cell death as well as heightened cell competition in mosaics, and that Xrp1 and its partner Irbp18 are involved (Baumgartner 2021 and Recausins 2021). The work of Igaki and his colleagues work adds to this and represents an important advance to the field, as it describes additional mutations that, along with the Rp+/- group, appear to form a common mechanism underlying cell competition in mosaic tissues.

Comments:

1. Although the very recently published papers from the Piddini and Vincent labs (Baumgartner 2021 and Recausins 2021) are not cited here (but should be), given those lab’s results that Rp+/- cells exhibit severe proteotoxic stress even in the absence of cell competition yet survive to form an animal after a developmental delay, I wonder if wol mutants show similar phenotypes: e.g, if the mutant tissues/animals were viable but developmentally delayed with sporadic cell autonomous cell death in the absence of cell competition. It would be enlightening to look at eye or wing development for timing and for cell death during the larval growth period, if the wol-/- mutants are viable (or if heterozygous wol mutants show increased ER stress).

2. Related to the thoughts in #1, the entirely wol mutant eye shown in Fig. 1 is smaller and looks like it has a reduced number of ommatidia, although they are regularly arrayed. Do the disc cells express Xbp1 or other markers that would indicate that they suffer ER stress? The way the wol mutant eyes were generated for Fig. 1 was via mosaicism coupled with WT-linked cell-lethality, to eliminate all WT cells from the eye disc. Since the eye seems slightly smaller than normal, could it be that compensatory growth is prevented by the absence of WT cells? If so, is this accompanied by a lack of developmental delay? Either way, such information on cell-autonomous cell death and developmental delay in the mutants would be very informative for the field.

3. Are Calr and Elp3 mutants also viable (+/- and/or -/-)?

4. Since not every context of cell competition was tested the title to Supp Fig. 1 seems a bit overstated. A suggesting is to change it to “ER stress underlies several different genetic contexts of cell competition”.

5. It is possible that the GCN2 RNAi is weak even with the addition of Dicer-2, thus yielding no suppression of competitive elimination of wol mutant cells. Do the authors know how efficient the GCN2 RNAi is in vivo?

5. In Supp Fig. 3 the label Xbp1s-GFP should be corrected to Xbp1-GFP

**Have all data underlying the figures and results presented in the manuscript been provided?**

Reviewer #1: None

Reviewer #2: Yes

PLOS authors have the option to publish the peer review history of their article (what does this mean?). If published, this will include your full peer review and any attached files.

Reviewer #1: No

Reviewer #2: **Yes: **Laura Johnston

---

## [Decision Letter · Decision Letter 1]

19 Nov 2021

Dear Dr Igaki,

We are pleased to inform you that your manuscript entitled "Cell competition is driven by Xrp1-mediated phosphorylation of eukaryotic initiation factor 2α" has been editorially accepted for publication in PLOS Genetics. Congratulations!

Yours sincerely,

Norbert Perrimon

Associate Editor

PLOS Genetics

Gregory P. Copenhaver

Editor-in-Chief

PLOS Genetics

Comments from the reviewers (if applicable):

Reviewer's Responses to Questions

**Comments to the Authors:**

Reviewer #1: The authors answered to all comments and their additional experiments solidified their claims. Their work is interesting for the cell competition field by adding a new system of cell competition as well as providing a convincing link between the proteotoxic stress and Xrp1. As I wrote before, I am convinced that this report will stimulate further work. All in all, the authors satisfactorily addressed most of my previous concerns with this revision.

Reviewer #2: The authors have addressed the reviewers comments. While I am still not in complete agreement with the interpretation that the ER stress response is a "mechanism of cell death" in these contexts of cell competition (as opposed to ER stress, not the response, as a mechanism leading to lower fitness), the data are important. It is a debate that will no doubt be continued in the literature.

**Have all data underlying the figures and results presented in the manuscript been provided?**

Reviewer #1: None

Reviewer #2: Yes

PLOS authors have the option to publish the peer review history of their article (what does this mean?). If published, this will include your full peer review and any attached files.

Reviewer #1: No

Reviewer #2: No

**Data Deposition**

http://datadryad.org/submit?journalID=pgenetics&manu=PGENETICS-D-21-00498R1

**Press Queries**

---

## [Editor Report · Acceptance letter]

30 Nov 2021

PGENETICS-D-21-00498R1 

Cell competition is driven by Xrp1-mediated phosphorylation of eukaryotic initiation factor 2α 

Dear Dr Igaki, 

We are pleased to inform you that your manuscript entitled "Cell competition is driven by Xrp1-mediated phosphorylation of eukaryotic initiation factor 2α" has been formally accepted for publication in PLOS Genetics! Your manuscript is now with our production department and you will be notified of the publication date in due course.

With kind regards,

Agnes Pap

PLOS Genetics

On behalf of:
